# Research on technology prospect risk of high-tech projects based on patent analysis

**Liwei Zhang** [1]*, **Zhihui Liu** [2]

**1** School of Management and Engineering, Capital University of Economics Business, Beijing, China,
**2** Research Center for Information Science Methodology, Institute of Scientific and Technical Information of China, Beijing, China

* zhangliwei19810@126.com

## Abstract

The uncertainty of high technology has determined the high-risk character of high-tech projects. Thus, it is of great importance to effectively avoid the risk of high-tech projects by thoroughly analyzing projects' methodologies and fully understanding the technology prospect risk of projects in the feasibility study phase. This study proposes a systematic research framework to identify and analyze the technology prospect risk of projects based on patent analysis. Thus, text mining technology and principal component analysis are used to improve the traditional patent map method and construct a technology prospect risk map. Moreover, patent value evaluation and correlation analysis methods are combined to identify technology potential areas and calculate the value of technology prospect risk. At the same time, an empirical study is conducted with project cases in the field of optical communications, and the technology prospect risk situations of these projects are ascertained through qualitative and quantitative methods. The study is innovative and practical and offers a better combination of analysis methods for current technology development and specific projects.

**Data Availability Statement:** All relevant data are within the manuscript and its Supporting Information files.

**Funding:** This work was supported by the Chinese National Social Science Foundation funded by the

## 1. Introduction

With the rapid development of the social economy, high technology plays a crucial role in the development of enterprises, countries, and the entire society. As the carriers of high technology, high-tech projects are based on technological innovation, and technical uncertainty can lead directly to project risk. Government services attempt to select valuable high-tech projects that would be beneficial to fund. Because specific technologies applied in projects play a very important role, they must be analyzed and evaluated before the projects are carried out to ensure that the upcoming projects possess good research and development (R&D) prospects. Technology prospect risk is mainly used to identify whether the technology used in a project has R&D value, whether there is a risk of infringement, and so on. Since patents are useful sources of knowledge about technical progress and innovative activity [1–6] and constitute reliable intelligence that reflects advances in technological development [7], patent analysis has

Chinese government (No.15CTQ031). The funders had no role in study design, data collection and analysis, decision to publish, or preparation of the manuscript.

**Competing interests:** The authors have declared that no competing interests exist.

been considered to be a vital tool for the formulation of technology strategy formulation and R&D planning [8].

Previous studies have identified technological directions or areas for technological opportunities from a broad view, yet few studies address how to detect and evaluate distinctive technologies that can act as new technological opportunities at the individual technology level. Previous studies on technological opportunities based on patent analysis have proposed approaches and systems for identifying potential technological types by exploiting keyword-based morphology analysis [9], for identifying undeveloped technological areas by analyzing patent vacuums [10], for forecasting products by temporal analysis of patent maps [11], and for generating discontinuous ideas of product improvements by exploiting system evolution patterns [12]. These previous studies focus on exploring technological opportunities that have not been explored, but few studies combine patent analysis with a specific project to judge the project's technology prospects.

To address the aforementioned problem, this paper proposes a method to evaluate the prospect risk of a specific technology based on patent analysis. The identification of technology prospect risk of a specific project can lead to the correct selection and increased investment of resources by the government, and speed up the overall research in specific fields. The analysis model outline is presented in Fig 1. The procedure consists of (1) collecting patent data, (2) determining keywords in a certain field, (3) transforming patent documents and the technological methods of a project into a keyword frequency matrix, (4) conducting principal component analysis, (5) constructing an initial technology prospect risk map of the project, (6) determining technology potential areas and technology barrier areas, and forming a technology prospect risk map of the project, (7) conducting correlation analysis and obtaining the technology prospect risk value of the project. By identifying the relationship and measuring the correlation degree between the technology procedure of a project and current patents in the same technology industry, the proposed method qualitatively and quantitatively analyzes the prospect risk of a project's technology procedure.

This paper is structured as follows. Section 2 addresses the underlying methodology for the proposed approach: patent analysis and patent map. Section 3 focuses on the overall research framework and details the processes for analyzing and evaluating the technology prospect risk of high-tech projects. A case study with an R&D project of automatic switching optical network node equipment, and optical communication technology-related patents is provided in Section 4, followed finally by the discussion and conclusion.

## 2. Related work

### 2.1 Patent analysis

Patent analysis is widely used in many countries, fields, industries, companies and technology sectors [13]. Patent literature presents an abundance of technical and market information, such as technical characteristics, ownership, and commercial worth [14, 15]. Approximately 80% of the world's technology knowledge [16, 17] can be found in patents. Patent documents are sourced from national patent offices, ensuring the authority and accuracy of patent data. Moreover, national patent offices have perfected the structure of patent documents, further enhancing the usability of patent data. Thus, researchers, including technology developers and policymakers, can easily access and utilize patents as data sources [18].

Patent documents are composed of structured and unstructured data: structured data have a consistent and standardized format, such as patent number, filing date, IPC classification number, patent family, cited patent, and patentee or inventors [10, 18, 19]; the unstructured patent data comprise narrative text including the patent title, abstract, claims, and description

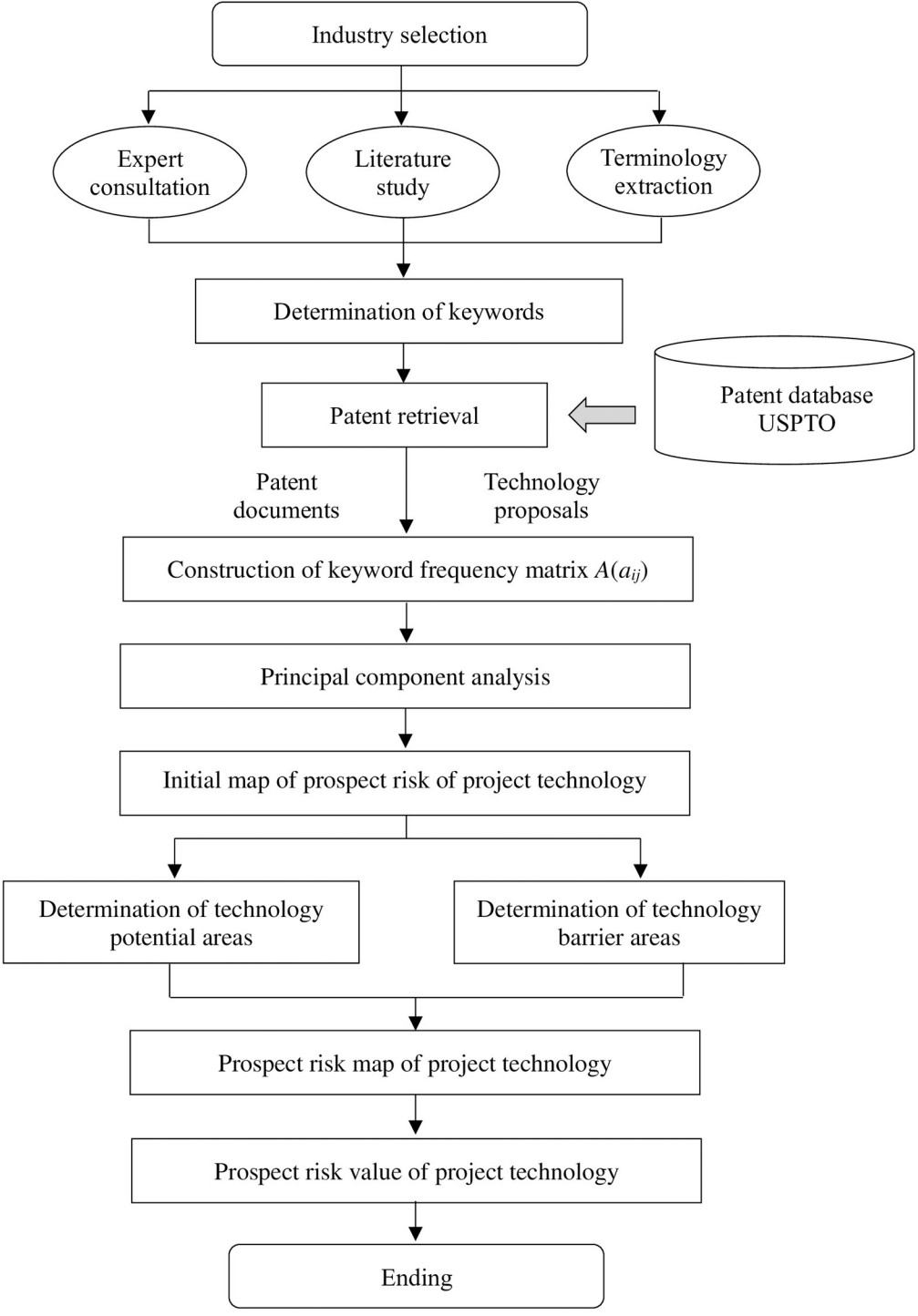

**Fig 1. Overall procedure for analyzing projects' technology prospect risks.**

[20, 21]. The methods and techniques for patent analysis are applied according to the characteristics of the structured and unstructured data. Therefore, patents can be analyzed from two viewpoints: quantitative and qualitative analysis. For the quantitative analysis method, patent analysis is applied to trend analysis, time-series analysis, and share analysis in accordance with

patent type, technological classification, assignees, and so on. The result of quantitative analysis has been shown in the form of numerical trends or curves [22, 23]. A bibliometric technique is used to investigate dynamic trends in technology development, and statistical methods make structured data more informative and constructive [9]. Social network analysis is increasingly popular as an advanced method of structured data analysis for monitoring technological development and identifying technological competition, such as technology development path tracking [24, 25], the integration of interdisciplinary technologies [26], and patent infringement lawsuits [27]. To conduct qualitative analysis, the contents of patent documents are utilized. Since it is difficult to systematically extract valuable information from natural languages, such as texts or descriptions, unstructured data is transformed into structured data [13]. Thus, text mining is a widely used technique that logically and automatically derives keywords from collections of unstructured data [28]. The keywords extracted from patents usually include technology, products, components, and functions [13]. The visual output of the structured patent data is represented in the form of graphs and networks, whereas the results from the unstructured patent data are represented as patent maps [29].

The purpose of patent analysis is diverse in terms of technical and economic decision making: identifying technological competitors and partners [30, 31], analyzing technology trends [32–35], identifying patent infringements [36, 37], conducting strategic technology planning [38–40], identifying patent quality [41–43], and determining novelty in patents [44, 45]. In fact, a technology strategy in firms specializing in technology acquisition, technology transfer, and even mergers and acquisitions can be formulated with the use of the methodological patent analysis described above [46].

## 2.2 Patent map

One patent analysis tool is a patent map that visualizes all the relationships among patents for a given technology. Patent maps showing competition trends in technological development can provide valuable input for decision support of R&D strategies [33]. A patent map is a visual representation of the relationship among patents and contains a variety of visual concepts and descriptions, such as charts, graphs, bar charts, and tables [47, 48], and significant amounts of technological information can be acquired in informative and easy ways. Thus, the patent map plays an important role in formulating strategies because it provides practical and intuitive information [49]. Because text mining technology can be applied to unstructured data processing to build patent maps, patent maps can effectively convey potentially valuable knowledge and explicit information from patents [21, 50]. Therefore, it is imperative to develop a patent map because it provides valuable information in a graphical manner, helping people to explore technological opportunities and avoid technological risks. Thus, research on patent maps has attracted the attention of interested people.

Because current technologies develop so rapidly and technology competitions become unusually fierce, it is imperative to avoid unnecessary investment and gain the seeds of technological development in the applicable fields. On the basis of such awareness, the Japan Patent Office has been producing and providing more than 50 types of expressions and more than 200 maps for several technology fields since 1997 [51, 52]. In addition, many other countries such as Korea, Italy [53, 54] and the USA [55] also provide many kinds of patent maps. Patent maps are categorized according to specific purposes: the technical patent map, the management patent map, and the claim patent map [9]. The technical map is used to understand core technology and identify potential technology. Management patent maps trace dynamic trends of specific technologies. Claim patent maps are useful for monitoring patent conflicts [13]. However, current patent maps have some drawbacks. According to Kim et al. [49], most of

them are time-based, ranking-based or matrix maps that consider only structured or unstructured items of patent documents.

## 3. Methods

This research applies text mining to analyze the unstructured items in patent documents, such as the titles, abstracts and claims, as well as the technical proposal of the high-tech project. Thus, the patents and the technical proposal are combined together to construct a keyword frequency matrix. Then, principal component analysis [10] and correlation analysis are used to evaluate the technology prospect risk of the project. At the same time, a visualization method is applied to develop the technology prospect risk map of projects, which can show the R&D value of the adopted technology and the technology barriers that may be touched on. The construction process is shown in Fig 1.

### 3.1 Determination of technical keywords and acquisition of patent documents

The study selected the research objective, located it in a specific technical area, determined the technical keywords, defined retrieval conditions and acquired the relevant patent documents.

1. Determination of technical keywords and definition of a patent retrieval query: The determination of keywords has a direct impact on the acquisition of patent documents. To improve the scientific nature and integrality of the keywords, keyword retrieval strategies in the relevant technology field are determined by three methods: terminology extraction [56], literature study and expert consultation. On the basis of these three means, the technical keywords of a specific industry are determined, and a patent retrieval query is defined.

2. Retrieval and acquisition of patent documents: The primary source of patents employed in this study is the United States Patent and Trademark Office (USPTO) database. The USPTO database is one of the most representative systems because patents submitted in the US are often simultaneously submitted in other countries, and the US has the largest commercial market in the world [57, 58]. The patent retrieval query is input to obtain patents from the USPTO patent database. Then, the patent documents are preprocessed—which requires cleaning, integration and loading—and clean data are obtained for the following data analysis.

### 3.2 Constructing a frequency matrix of keywords

**3.2.1 Feature representation of documents.** Feature representation refers to text information being represented by feature items. We only need to process the feature items to comprehend unstructured text processing. At present, there are a variety of feature representation models, such as the Boolean model, vector space model, and probabilistic model. In this paper, according to the characteristics of patent documents, the vector space model [59] is preferred.

In the vector space model, the patent documents are considered to be a set of vectors composed of feature items, and each document is expressed as a standardized eigenvector $V(D) = (t_1, w_1; t_2, w_2; \ldots ; t_n, w_n)$, where $t_i$ is a feature item, and $w_i$ is the weight of $t_i$ in the document D. In this way, all patent documents constitute a vector space. When the collection of patent documents is fixed, the position of $t_i$ is fixed as well; thus, the eigenvector can be simplified as $V(D) = (w_1, w_2, \ldots, w_n)$.

The weight value depends on the selection of the weighted formula, which directly affects the performance of text categorization. There are two kinds of frequently used weight

calculation methods: two-value representation, which means "if $t_i$ appears in the article, $w_i$ is 1, otherwise 0", where $t_i$ is a feature item, and $w_i$ is the weight of $t_i$; and frequency feature, which means that $w_i$ represents the appearance frequency of $t_i$, such as 0, 1, 2. The first method is simpler, but much useful information is lost. The second one is more reasonable, as it retains the frequencies of feature items. To obtain more accurate results, the frequency feature is chosen to calculate the weight of the feature items in this study.

**3.2.2 Forming a keyword frequency matrix.** In this section, we analyze the unstructured items in patent documents, including names, abstracts, claims and so on. At the same time, the technology adopted by the high-tech project must be expressed in the same format as the patent documents, such as names, abstracts and specifications, and studied together with the patent documents.

The research method is as follows: according to the identified keywords, the appearance frequency of the keywords in the names, abstracts and specifications are counted. The keyword frequency matrix $A(a_{ij})_{(p \times q)}$ is obtained and is defined as follows:

$$A(a_{ij})_{(p \times q)} = \begin{bmatrix} a_{11} & a_{12} & \cdots & a_{1q} \\ a_{21} & a_{22} & \cdots & a_{2q} \\ \cdots & \cdots & \ddots & \cdots \\ a_{p1} & a_{p2} & \cdots & a_{pq} \end{bmatrix}$$

where p is the number of documents (patents and projects), q is the number of keywords, and $a_{ij}$ is the frequency of the j-th keyword appearing in the i-th document.

**3.2.3 Initial map of projects' technology prospect risk.** A given industry includes many technical keywords, and one document usually contains more than 10 keywords. Thus, it is difficult to grasp the characteristics of each document, let alone the technology layout of the entire industry. To address this issue, principal component analysis is used to analyze the keyword frequency matrix and obtain its principal components. Then, a coordinate system is built based on the principal components, and the patents and project information are mapped onto a multidimensional space. Finally, we construct an initial map of the projects' technology prospect risk. The development process is shown in Fig 2.

**3.2.4 Determination of technology potential areas.** Fig 2 shows the patent layout in one particular industry. The entire space represents one specific technical field, and there are many vacant areas that correspond to undiscovered technical areas. However, whether these areas are worth being developed remains to be analyzed.

1. Setting the technology potential areas.
   First, one or more relatively large areas with lower patent density need to be determined. The rule for selecting potential areas is that the patent density is relatively sparse but not too low. If the patent density is too high, the area has been the focus of R&D and has many technology barriers, with little R&D potential. If the density is too low, few studies have been performed in the area. In this case, the risk would be very high if R&D is carried out in the area. Because the type of projects to be carried out in this paper focuses on the application of a technology's R&D rather than basic theory research, an area with very low density is not suitable. First, because any application technology's R&D is based on the basic theory research, it is necessary that the basic theory research has matured; second, even if the theoretical foundation has matured, it is still unknown whether the technology can be industrialized, as nobody is trying to do so and few experiences can be referred to. Thus, this is an

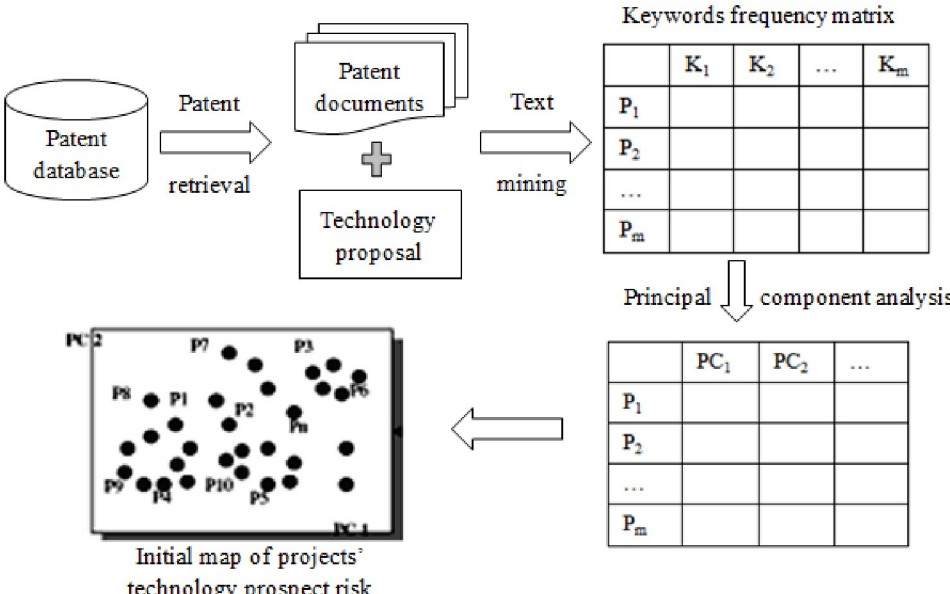

**Fig 2. Constructing process for initial map of projects' technology prospect risk.**

extremely high-risk area. If a project in this area is conducted blindly, its R&D progress is full of risk.

According to the principles above, the possible technology areas of potential are established. After developing the initial map of technology prospect risk and identifying the patent vacancy areas, the next step is to verify the validity of each vacancy area.

2. Verifying technology potential areas.

Some vacancy areas seem valuable, but due to the low potential value of surrounding patents, the vacancy areas are in fact barren. Therefore, it is indispensable to verify the effectiveness of the technology potential areas. The standards of identifying a valuable vacant area depend on the significance of its surrounding patents.

The process of verifying the technology potential areas is as follows: First, the original information of all surrounding patents is collected and used to define vacancy areas. The range of patent information is broad and diverse, from basic information (patent numbers, patent names, etc.), to more complex information, such as abstracts, claims and so on. Secondly, the evaluation criteria of the technology potential area's effectiveness should be set up. Through expert interviews and literature reviews, three kinds of indicators are established:

a. Importance indicators:

- Number of patent claims. The number of patent claims can indicate the scope or width of patents [60], and show the value of patents. A previous study [61] showed that a more valuable patent would invite more claims, cover a wider technology range, and encounter more infringement charges and litigations. This study uses the average value of claims requested by surrounding patents to represent the number of patent claims.

- Patent family. Patent family is defined as "the set of patents (or applications) filed in several countries which are related to each other by one or several common priority

filings" by the OECD (Organization for Economic Co-operation and Development) Patent Statistics Manual [62]. Given the territorial character of patent protection, when applicants want to protect their inventions in different countries, a patent application needs to be filed in each one of the patent offices where protection is sought. As a result, the first patent filing made to protect the invention (the priority filing) is followed by a series of subsequent filings and together they form a patent family [63]. Because the application fee and annual maintenance fee of patents are very high, a patent whose patent family includes more members is more valuable.

b. Novelty of technology: this factor is reflected by the characteristic coefficient of the new technology [64].

- Characteristic coefficient of the new technology n: $n = \sqrt{v^2 + \alpha^2}$, where v is the growth rate of the technology and α is the maturity coefficient of the technology. It is a comprehensive indicator to reflect the degree of emergence or aging of the technology. A larger n means that its new technology features are more obvious and that it possesses more development potential.

- Growth rate of technology: v = a/A, where a is the application quantity (or approval quantity) of the inventions in one year, and A is the cumulative application quantity (or cumulative approval quantity) of the inventions dating back 5 years. Calculated for several years in a row, the value of v is increasing, showing that the technology is in the inception or growth stage.

- Mature coefficient of technology: $\alpha = a/(a + b)$, where a is the application quantity (or approval quantity) of the inventions in one year, and b is the application quantity (or approval quantity) of the utility models in one year. Calculated for several years in a row, the value of α is decreasing, showing that the technology is becoming increasingly mature.

c. Infringement risk: this risk is evaluated by the patent licensing rate.
Based on the above indicators, the technology potential areas can finally be determined.

**3.2.5 Determination of the technology barrier areas.** The determination of the technology barrier areas is easier than that of the technology potential areas. The technology barrier areas are selected based on the patent density, and the densest one is usually chosen as the technology barrier area. The technology barrier area could be identified according to the patent distribution on the initial map of technology prospect risk.

**3.2.6 Technology prospect risk map of projects.** Technology potential areas are determined according to the validity test, while technology barrier areas are identified by referring to the patent density. Then, the technology prospect risk areas of the project are defined, and the technology prospect risk map is constructed. Finally, the technology prospect risk of the project can be qualitatively judged according to the position of the technology in the risk map.

**3.2.7 Analysis of projects' technology prospect risk.** The technologies adopted in high-tech projects, the patents surrounding the technology potential areas (PP) and the patents surrounding the technology barrier areas (PB) are separately expressed as keyword frequency matrixes TS, TP, and TB. Then, the correlation analysis between TS and TP, as well as the correlation analysis between TS and TB, are both carried out. Finally, the value of technology prospect risk can be obtained. The specific method is as follows:

In this paper, the similarity degree [59] is applied to measure the correlation between documents $D_t$ and $D_p$. $D_t$ represents the keyword frequency vector of the technology adopted by a

high-tech project, and $D_p$ represents the keyword frequency vector of a patent document. The similarity degree is defined as the distance between $D_t$ and $D_p$ and expressed is by the angle cosine formula:

$$Sim\left(D_t, D_p\right) = cos(\theta) = \frac{\sum_{k=1}^{n} w_{tk} w_{pk}}{\sqrt{\sum_{k=1}^{n} w_{tk}^2} \times \sqrt{\sum_{k=1}^{n} w_{pk}^2}} \tag{1}$$

where $D_t = (w_{t1}, w_{t2}, \ldots, w_{tn})$, $D_p = (w_{p1}, w_{p2}, \ldots, w_{pn})$, and w represents the frequency of a certain keyword appearing in a certain document.

The similarity degree between the technologies adopted by high-tech projects and patents in the same industry can be revealed by calculation. Suppose the similarity degree between the technologies and PP is $S_{tp}$, and the similarity degree between the technologies and PB is $S_{tb}$. According to the analysis above, the effective technology potential area represents the low-risk area, while the technology barrier area is in the high-risk area; thus,

$$PR = S_{tp} \odot [(-1) * S_{tb}] \tag{2}$$

where $\odot$ represents the synthesis between $S_{tp}$ and $S_{tb}$. Aiming at a specific research object, the specific synthesis method is adopted through many experiments.

## 4. Empirical study

With the continuous exponential growth of IP services, the reform of the telecommunications regulatory system, and the gradual and comprehensive opening of the telecommunications market, optical communications will gradually replace traditional switching, transmission, and access technologies, and ultimately achieve all-optical networking. In view of the obvious advantages and development prospects of optical transmission networking, many countries have invested a lot of manpower, material and financial resources in pre-research. The US Department of Defense Research Agency (DARPA) has funded a series of optical networking projects, such as "Optical Network Technology Cooperation Program (ONTC)" developed mainly by Bellcore,"All-Optical Communication Network" pre-research program developed mainly by Lucent, and so on. In Europe and Japan, similar optical transmission networking projects have also been carried out. At the same time, China has also arranged a cross-themed Chinese high-speed information demonstration network project in the national 863 high-tech plan. Obviously, optical transmission networking has become another new global optical communication development climax after SDH electrical networking.

An R&D project with automatically switched optical network (ASON) node equipment in the optical communication field——"ASON Automatic Switching Optical Network Node Equipment Development and System"is selected for the empirical study. The project is to finish the R&D tasks of ASON with functions including multiple types of business access and dynamic resource allocation. As the undertaking unit of the project, P Company relying on its leading technical strength and deep understanding of optical network requirements, is actively committed to the research and development of ASON equipment and how the existing traditional network can smoothly evolve to the next-generation intelligent optical network ASON. The paper analyzes the technology prospect risk of the project accordingly.

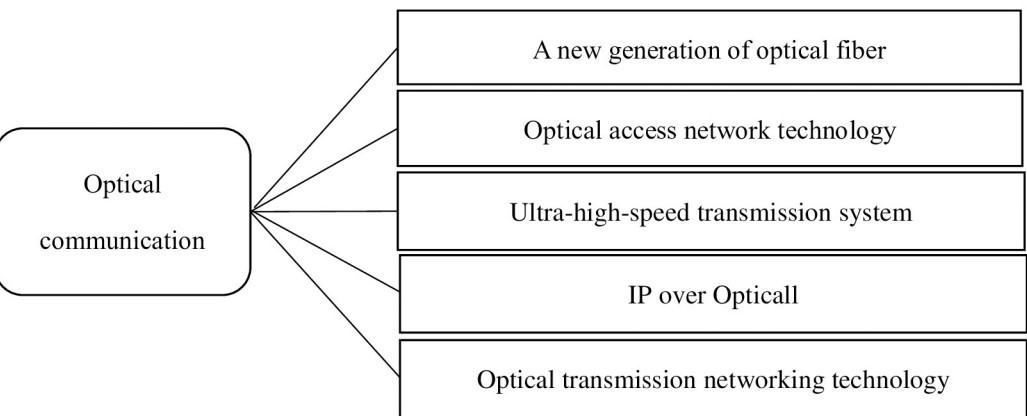

**Fig 3. The taxonomy of optical communication.**

## 4.1 Data collection

At present, the five development directions in the field of optical communications are a new generation of optical fiber, optical access network technology, ultra-high-speed transmission systems, IP over Opticall, and optical transmission networking technology, which is shown in Fig 3.

Through expert consultation and literature review, the patent retrieval query for each sub-field of the Optical Communication field were determined (Table 1). For analysis, we applied the keywords in Table 1 and collected the patents issued in the USPTO from 2000 to 2020. The total number of patents is 21002, and the detail is displayed in the Table 1.

**Table 1. Patent retrieval query of optical communication.**

| Technical Field | Sub-Technical Field | Patent Retrieval Query | Total Number Of Patents |
|---|---|---|---|
| Optical Communication | A New Generation Of Optical Fiber | TA_ALL((optical transmission OR WDM OR optical communication OR Fibre communication) AND (transmission fiber OR modulation format OR broad band optical amplification OR dispersion compensation OR polarization mode dispersion compensation OR dispersion management OR coherent detection OR balance detection OR electrical equalization OR gain equalization OR forward error correction))AND APD:[20000101 TO 20200630] | 15541 |
| | Optical Access Network Technology | TA_ALL((optical access OR optical network OR optical communication) AND (burst mode transmitter OR burst mode receiver OR media access control OR bandwidth allocation OR security authentication OR ranging OR protection OR restoration))AND APD:[20000101 TO 20200630] | 832 |
| | Ultra-High-Speed Transmission System | TA_ALL((optical switching OR optical burst switching OR optical communication OR multicast) AND (optical switch OR optical circuit switching OR optical packet switching OR optical switching fabric OR optical add/drop multiplexing OR reconfigurable OADM OR multi-granularity optical switching OR scheduling OR contention resolution OR tunable)) AND APD:[20000101 TO 20200630] | 2021 |
| | IP Over Opticall | TA_ALL((NGN OR soft switching OR ASON) AND (security OR server OR service aware OR multimedia application OR manageable OR controllable OR architecture OR network measurement OR dual stack OR codec OR multicast OR P2P))AND APD:[20000101 TO 20200630] | 451 |
| | Optical Transmission Networking Technology | TA_ALL((optical networking OR optical network OR optical communication) AND (control plane OR protection OR restoration OR multiple protocol label switching OR routing OR signaling OR GMPLS OR optical interconnect OR contention resolution OR layering OR Cross-Domain OR UNI OR NNI OR interface))AND APD:[20000101 TO 20200630] | 2157 |

**Table 2. Principal component analysis table of keywords.**

| | Principal components | | | | |
|---|---|---|---|---|---|
| | **1** | **2** | **3** | **4** | **5** |
| optical transmission | 0.268 | -0.327 | 0.406 | -0.136 | 0.337 |
| wavelength division multiplexing | 0.395 | -7.66E-002 | 0.214 | -6.38E-003 | 0.310 |
| optical communication | 0.197 | 1.40E-002 | 0.217 | 0.253 | -0.332 |
| transmission fiber | 0.180 | -0.161 | 0.255 | -5.92E-002 | 0.206 |
| modulation coding | 2.84E-002 | -1.40E-002 | 4.13E-002 | 4.45E-002 | -0.138 |
| dispersion compensation | 0.295 | -0.301 | 0.398 | -0.152 | 0.372 |
| polarization mode dispersion compensation | 7.34E-002 | -4.19E-002 | 6.72E-002 | -1.82E-002 | 0.113 |
| dispersion management | 3.67E-002 | -6.49E-002 | 7.19E-002 | -3.74E-002 | 6.46 E-002 |
| coherent reception | 6.52E-002 | -3.1E-002 | 5.16E-002 | -1.04E-002 | 8.55 E-002 |
| balanced reception | 3.67E-002 | -6.49E-002 | 7.19E-002 | -3.74E-002 | 6.46 E-002 |
| electronic equalization | 1.71E-002 | -1.5–002 | 2.38E-002 | 1.79 E-002 | -8.77 E-002 |
| gain equalization | 7.43E-002 | -6.22E-002 | 9.26E-002 | 9.55 E-003 | -4.73 E-002 |
| forward error correction | 2.13E-002 | -4.05E-002 | 3.92E-002 | -2.30E-002 | 1.36 E-002 |
| optical access | -2.32E-002 | -4.18E-003 | 5.72E-002 | 0.101 | 1.12 E-003 |
| optical network | -0.567 | 8.99E-002 | -0.165 | 5.20 E-002 | 6.10 E-002 |
| medium access control | -0.19 | 0.348 | 0.35 | 3.37 E-002 | 7.77 E-003 |
| bandwidth allocation | -0.145 | -6.33E-002 | -4.39E-002 | 0.106 | -0.168 |
| ranging | -6.14E-002 | -2.32E-002 | -1.00E-002 | 2.91 E-002 | -0.145 |
| …… | …… | …… | …… | …… | …… |
| …… | …… | …… | …… | …… | …… |

## 4.2 Initial map formation of projects' technology prospect risk

**4.2.1 Principal component analysis.** According to the techological solutions of the project and the patent data obtained in the field of optical communications, combined with the keywords in Table 1, a keyword word frequency matrix is constructed. According to the method proposed in section 3.2.3, perform principal component analysis on the generated word frequency matrix to obtain the component matrix, as shown in Table 2. There are a total of five factors whose eigenvalues are shown in Table 3.

Through analysis and verification, only factor 1 and factor 3 are effective. Therefore, factor 1 and factor 3 are selected to construct the coordinate system.

**4.2.2 Mapping patent documents.** Through the analysis in section 4.2.1, factor 1 and factor 3 act as the X-axis and Y-axis, respectively, to set up the coordinate system, and the patent documents are mapped onto two-dimensional space. The result is shown in Fig 4.

## 4.3 Setting up the initial technology potential areas

In this section, the initial technology potential areas are set up. According to Fig 5, the marked areas may be the technology potential areas, but further validation work is needed.

## 4.4 Validating technology potential areas

Through the effectiveness analysis, the situations of the selected six areas can be seen in Table 4.

**Table 3. Eigenvalue table of principal factors.**

| Factor number | Factor 1 | Factor 2 | Factor 3 | Factor 4 | Factor 5 |
|---|---|---|---|---|---|
| Eigenvalue | 0.076 | -0.183 | 0.070 | -0.081 | -0.277 |

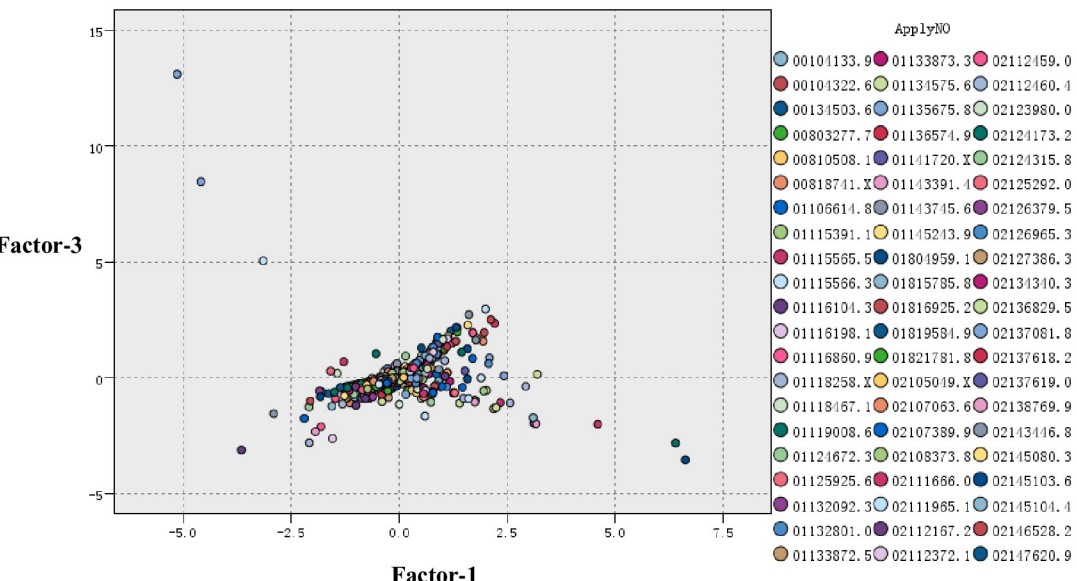

**Fig 4. Initial map formation of projects' technology prospect risk.**

According to the research results in section 3.2.4, there are four rules that can identify the technology potential areas:

1. More claims represent better patent quality. The areas surrounded by patents with more claims possess better development prospects.

2. Lower authorized rates mean less infringement possibilities. The areas surrounded by patents with lower authorization rates possess larger development space.

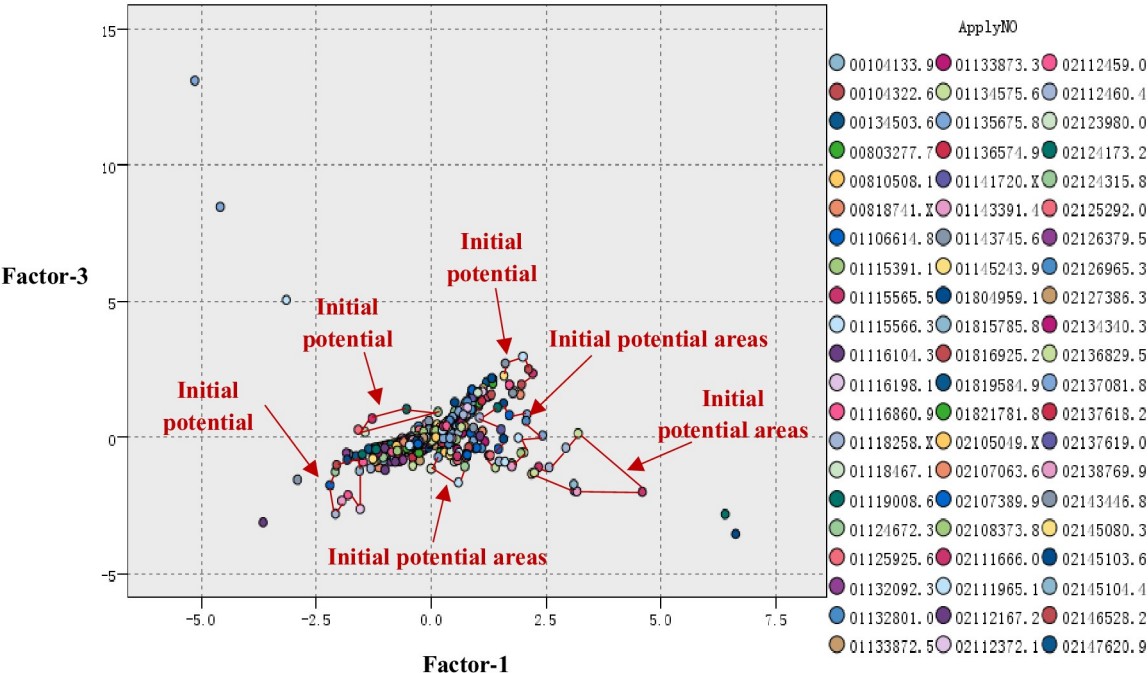

**Fig 5. Initial potential areas.**

**Table 4. Indicators table of potential areas.**

| GROUP | Claims number | Authorized rate | Feature coefficient of new technology | Patent families |
|---|---|---|---|---|
| GROUP1 | 5.30 | 30.0% | 0.30 | 4 |
| GROUP2 | 13.40 | 40.0% | 0.60 | 10 |
| GROUP3 | 7.25 | 50.0% | 0.25 | 3 |
| GROUP4 | 13.20 | 20.0% | 0.67 | 7 |
| GROUP5 | 6.80 | 40.0% | 0.50 | 5 |
| GROUP6 | 7.64 | 35.7% | 0.36 | 3 |

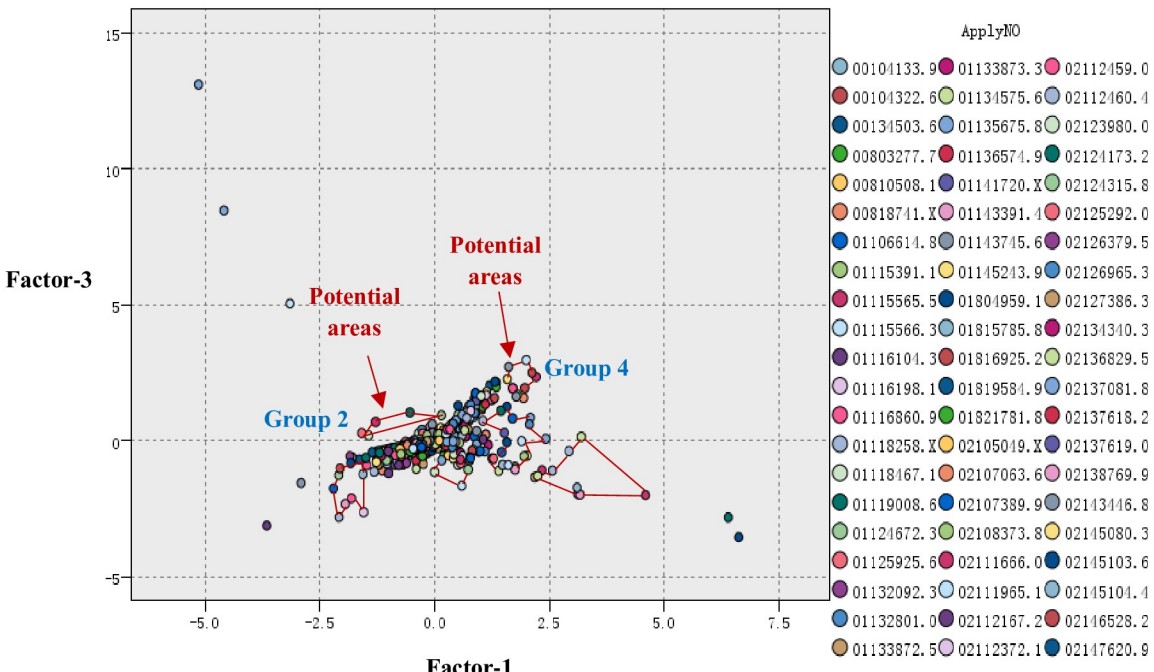

**Fig 6. Technology potential areas.**

3. Larger characteristic coefficients of new technology indicate stronger new technical features and more development potential.

4. Larger patent families reflect better patent quality. The areas surrounded by patents whose patent families include more members possess more development potential.

Therefore, considering these four rules, GROUP2 and GROUP4 are the first choices for R&D potential regions, as shown in Fig 6. Table 5 shows some patents surrounding the two regions.

**Table 5. Surrounding patent table of effective technology potential areas.**

| Application Number | Patent Name | Applicant | Patent Claims | Authorized | Applying Time | Patent family |
|---|---|---|---|---|---|---|
| US13/221792 | Passive optical network system | HITACHI, LTD. | 13 | authorized | 20110830 | 4 |
| US11/622181 | Method and system for compensating for optical dispersion in an optical signal in a hybrid optical network | FUJITSU LIMITED | 21 | authorized | 20070111 | 2 |
| US13/682705 | Equalization delay agnostic protection switching in protected passive optical networks | ZTE CORPORATION ZTE (USA) INC. | 19 | authorized | 20121120 | 13 |

## 4.5 Technology prospect risk map of project

Based on the above analysis and combined with patent density analysis, the technology potential areas and the technology barrier areas are displayed in Fig 7.

## 4.6 Technology prospect risk of project

According to the optical communication industry to which the study objects belong, a large number of projects in the field are selected and verified, and the formula for calculating the technology prospect risk is obtained, as shown in formula (3):

$$PR = S_{tp} + [(-1) * S_{tb}] \tag{3}$$

According to the correlation analysis among the high-tech project, PP and PB, and combined with the calculation of formula (3), the risk value of the technology PR is equal to 0.451, which means $S_{tp} > S_{tb}$. Thus, we know that the technology prospect risk of the project is less.

The validity of the experimental results can be verified by the actual implementation of the project. Through the implementation of the "ASON Automatic Switching Optical Network Node Equipment Development and System" project, P Company has successfully developed ASON node equipment with T-bit switching capacity, which has been applied to the National High-tech Plan Project "High-performance Broadband Information Network" and many other information networks. At the same time, P Company has also launched the converged intelligent optical network solution to provide a full range of support for the evolution of operators' existing transmission networks to the next generation of intelligent optical networks.

The project has achieved innovative results in the ASON node equipment frame structure, large cross-capacity non-blocking cross-over matrix construction and so on, which have been applied for and authorized more than 20 invention patents. Moreover, P Company has also participated in the formulation and revision of a number of ASON-related international

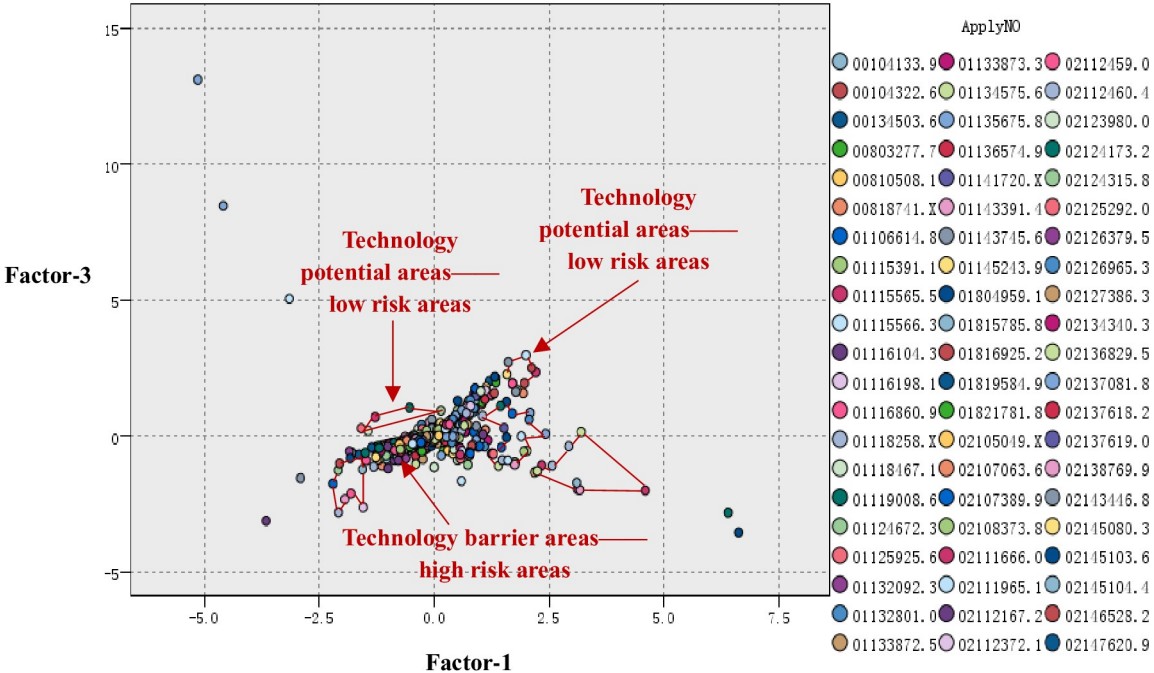

**Fig 7. Technology prospect risk map of project.**

standards. In addition, P Company has not been subject to patent infringement lawsuits for any of the final results.

## 5. Discussion and conclusion

At present, government services attempt to select some promising high-tech projects to fund, but high-tech projects commonly have high technology risk. Thus, it is imperative to evaluate the technology potentials and risks before the projects start. To address this problem, the paper proposed an approach to evaluate the technology prospect risk of high-tech projects based on systematic processes and scientific methods——to construct a technology prospect risk map and calculate the technology prospect risk value.

The proposed approach has several advantages over previous methods. Firstly, most previous studies focus on exploring technological opportunities that have not yet been explored, so their outputs provide a broad view of technological directions or technological areas that are promising and undeveloped [1]. However, few studies focus on a specific technology to explore whether it can act as a new technological opportunity at the individual technology level. The paper proposes a method for exploring the potential of a specific technology and evaluating the risk of that technology. According to the empirical research, this approach is significant and offers valuable practical guidance in selecting R&D projects, as it combines the information about technology development in a certain industry with a specific project application. Secondly, previous researches pay more attention to the identification of patent vacuums, while the evaluation of technology potential and risk takes into account not only technology potential areas, but also technology barrier areas. The research makes full use of the patent information around technology potential areas and technology barrier areas, and applies text mining, visualization methods and the correlation analysis method to construct a technology prospect risk map, as well as to calculate the value of the technology prospect risk, so as to synthetically evaluate the technology prospect risk. Finally, in order to make the research results more practical, we have developed a software system to implement our method more simply and efficiently, reducing the burden of manual work and therefore allowing even those who are unfamiliar with the complex algorithms to benefit from the research results. Then managers, researchers and engineers interested in new technology development can effectively evaluate project technology prospects and avoid technology risks before a project is approved and started. In addition, the analytical results can be updated easily with minimal involvement of experts, since the patent documents are totally reusable. As time goes on, new patent data and new technical terms will be separately supplemented into the original patent database and keywords set. When new projects need be evaluated, the projects will act as inputs, and their technology prospect risk will be outputs. Thus, this approach ensures the effectiveness and reusability of the software system.

Despite its contributions, this study exists only at the explorative stage and is subject to certain limitations as outlined below. Firstly, it is vital to elaborate the keyword extraction process, since keyword extraction plays a critical role in analyzing the technology prospect risk. Although the term extraction algorithm and expert judgment are employed to extract the keywords, covering both the quantitative and qualitative aspects of keyword extraction, other systematic methodologies should supplement these techniques in order to validate the extracted keywords. Secondly, to avoid identifying the technology potential areas subjectively, the study selects three kinds of indicators to evaluate the technology potential areas. Future research could integrate the existing body of literature on value and potential assessment of individual patents and technologies with the proposed approach to construct a systematic evaluation system of technology potential areas.

Nevertheless, the research opens up a new approach for the application of patent map by combining patent information with risk assessment of high-tech projects closely. At the same time, the development of a technology prospect risk map helps people to learn about the R&D potential of a project and technology barriers that may be encountered in the R&D process. We expect that the proposed method can be incorporated into the strategic technology planning process to assist technology experts in identifying project technology risks before a project is funded.

## Supporting information

**S1 Dataset.**
(XLS)

## Acknowledgments

We are grateful for the support and technical assistance from Prof. Shao Shicai at Ministry of Science and Technology the People's Republic of China.

## Author Contributions

**Conceptualization:** Liwei Zhang.

**Data curation:** Liwei Zhang, Zhihui Liu.

**Formal analysis:** Liwei Zhang.

**Investigation:** Liwei Zhang.

**Methodology:** Liwei Zhang.

**Resources:** Liwei Zhang.

**Validation:** Zhihui Liu.

**Writing – original draft:** Liwei Zhang.

**Writing – review & editing:** Liwei Zhang, Zhihui Liu.

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
