## [Decision Letter · Decision Letter 0]

20 Apr 2020

PONE-D-20-04089

Research on technology prospect risk of high-tech projects based on patent analysis

PLOS ONE

Dear Prof Zhang,

Thank you for submitting your manuscript to PLOS ONE. After careful consideration, we feel that it has merit but does not fully meet PLOS ONE’s publication criteria as it currently stands. Therefore, we invite you to submit a revised version of the manuscript that addresses the points raised during the review process.

The manuscript required further revisions regarding particularly empirical approach and discussion, alongside contribution to the related literature.

We would appreciate receiving your revised manuscript by Jun 04 2020 11:59PM. To enhance the reproducibility of your results, we recommend that if applicable you deposit your laboratory protocols in protocols.io, where a protocol can be assigned its own identifier (DOI) such that it can be cited independently in the future. For instructions see: http://journals.plos.org/plosone/s/submission-guidelines#loc-laboratory-protocols

We look forward to receiving your revised manuscript.

Kind regards,

Stefan Cristian Gherghina, PhD. Habil.

Academic Editor

PLOS ONE

Journal Requirements:

2. Thank you for stating the following in the Acknowledgments Section of your manuscript: "This work was supported by Chinese National Social Science Foundation funded by the Chinese government (No.15CTQ031)."

Please remove any funding-related text from the manuscript and let us know how you would like to update your Funding Statement. Currently, your Funding Statement reads as follows: "No"

Reviewers' comments:

Reviewer's Responses to Questions

**Comments to the Author**

1. Is the manuscript technically sound, and do the data support the conclusions?

Reviewer #1: Yes

Reviewer #2: Partly

2. Has the statistical analysis been performed appropriately and rigorously? 

Reviewer #1: Yes

Reviewer #2: N/A

3. Have the authors made all data underlying the findings in their manuscript fully available?

Reviewer #1: Yes

Reviewer #2: Yes

4. Is the manuscript presented in an intelligible fashion and written in standard English?

Reviewer #1: Yes

Reviewer #2: Yes

5. Review Comments to the Author

Reviewer #1: The paper investigates an interesting topic. I would suggest to expand the discussion or to integrate the discussion with conclusion in one section (to avoid many short sections), and to emphasize also contribution to the knowledge (in terms of extension of past studies). Author might also make a note to the usage of patent analysis in different sectors (from marketing to material science and so on).

I would also suggest to use image with a better resolution.

Reviewer #2: The authors propose a systematic research framework based on patent analysis to identify and analyze the prospective risk of high technology.

The introduction is clearly written and explains well the problem and the motivation for this work. The related work has been largely addressed with a sufficient number of quotations. The methodology is explained in sufficient detail. However, the empirical study is lacking in detail, especially in providing a background of the application field (optical communication). Moreover, it is complicated to understand how the keywords were found. The search query used for patent extraction from the USPTO database is missing. The process of determination of technical keywords and acquisition of patent in empirical study, needs to be addressed. it would be good to describe all stages of framework application in more details during the empirical study.

The discussion should also focus on the results of the patent analysis carried out during the empirical study in order to provide evidence on the effectiveness of the presented framework.

6. PLOS authors have the option to publish the peer review history of their article (what does this mean?). If published, this will include your full peer review and any attached files.

Reviewer #1: No

Reviewer #2: No

---

## [Author Response · Author response to Decision Letter 0]

3 Sep 2020

Thank you very much for your comments for our manuscript.

Reviewer #1

Comment 1: The paper investigates an interesting topic. I would suggest to expand the discussion or to integrate the discussion with conclusion in one section (to avoid many short sections), and to emphasize also contribution to the knowledge (in terms of extension of past studies). Author might also make a note to the usage of patent analysis in different sectors (from marketing to material science and so on).

Response: According to the Reviewer's suggestion, we have integrate the discussion with conclusion in one section. At the same time, we emphasize contribution to the knowledge. You can see “5. Discussion and Conclusion section”.

Comment 2: I would also suggest to use image with a better resolution.

Response: As the reviewer’s suggestion, we revised our figures. You can see the figure 4-7.

Reviewer #2

Comment 1: The introduction is clearly written and explains well the problem and the motivation for this work. The related work has been largely addressed with a sufficient number of quotations. The methodology is explained in sufficient detail. However, the empirical study is lacking in detail, especially in providing a background of the application field (optical communication). 

Response: According to the Reviewer's suggestion, we added background of the application in our manuscript. You can see the line 284-305.

Comment 2: Moreover, it is complicated to understand how the keywords were found. The search query used for patent extraction from the USPTO database is missing. The process of determination of technical keywords and acquisition of patent in empirical study, needs to be addressed.

Response: As the reviewer’s suggestion, we added the 4.1 Data collection section, Table 1 and Figure 3.

Comment 3: It would be good to describe all stages of framework application in more details during the empirical study.

Response: Thank you for reviewer’s suggestion. We have described all stages of the framework application in the methods section. Our purpose is that the relevant personnel can repeat our work in the future. We hope to provide an idea and method to reduce the technology prospect risk for high-tech enterprises. Therefore, we described the important steps in the empirical study section and showed the important results. This will make the structure of our manuscript more reasonable to suit the requirements of PLoS One journal. 

Comment 4: The discussion should also focus on the results of the patent analysis carried out during the empirical study in order to provide evidence on the effectiveness of the presented framework.

Response: we revised this section. You can see line 355-367. The validity of the experimental results can be verified by the actual implementation of the project. Through the implementation of the "ASON Automatic Switching Optical Network Node Equipment Development and System" project, P Company has successfully developed ASON node equipment with T-bit switching capacity, which has been applied to the National High-tech Plan Project "High-performance Broadband Information Network" and many other information networks.At the same time,P Company has also launched the converged intelligent optical network solution to provide a full range of support for the evolution of operators' existing transmission networks to the next generation of intelligent optical networks.

The project has achieved innovative results in the ASON node equipment frame structure, large cross-capacity non-blocking cross-over matrix construction and so on,which have been applied for and authorized more than 20 invention patents. Moreover, P Company has also participated in the formulation and revision of a number of ASON-related international standards. In addition, P Company has not been subject to patent infringement lawsuits for any of the final results.

---

## [Decision Letter · Decision Letter 1]

18 Sep 2020

Research on technology prospect risk of high-tech projects based on patent analysis

PONE-D-20-04089R1

Dear Dr. Zhang,

We’re pleased to inform you that your manuscript has been judged scientifically suitable for publication and will be formally accepted for publication once it meets all outstanding technical requirements.

Kind regards,

Stefan Cristian Gherghina, PhD. Habil.

Academic Editor

PLOS ONE

Additional Editor Comments (optional):

Reviewers' comments:

Reviewer's Responses to Questions

**Comments to the Author**

1. If the authors have adequately addressed your comments raised in a previous round of review and you feel that this manuscript is now acceptable for publication, you may indicate that here to bypass the “Comments to the Author” section, enter your conflict of interest statement in the “Confidential to Editor” section, and submit your "Accept" recommendation.

Reviewer #1: All comments have been addressed

Reviewer #2: All comments have been addressed

2. Is the manuscript technically sound, and do the data support the conclusions?

Reviewer #1: Yes

Reviewer #2: Yes

3. Has the statistical analysis been performed appropriately and rigorously? 

Reviewer #1: Yes

Reviewer #2: N/A

4. Have the authors made all data underlying the findings in their manuscript fully available?

Reviewer #1: Yes

Reviewer #2: No

5. Is the manuscript presented in an intelligible fashion and written in standard English?

Reviewer #1: Yes

Reviewer #2: Yes

6. Review Comments to the Author

Reviewer #1: I'm satisfied with the changed. I think that the current version of the paper can be accepted for publications with no further changes

Reviewer #2: (No Response)

7. PLOS authors have the option to publish the peer review history of their article (what does this mean?). If published, this will include your full peer review and any attached files.

Reviewer #1: No

Reviewer #2: No

---

## [Editor Report · Acceptance letter]

25 Sep 2020

PONE-D-20-04089R1 

Research on technology prospect risk of high-tech projectsbased on patent analysis 

Dear Dr. Zhang:

I'm pleased to inform you that your manuscript has been deemed suitable for publication in PLOS ONE. Congratulations! Your manuscript is now with our production department. 

Kind regards, 

on behalf of

Dr. Stefan Cristian Gherghina 

Academic Editor

PLOS ONE